# Drug Regimens of Novel Antibiotics in Critically Ill Patients with Varying Renal Functions: A Rapid Review

**DOI:** 10.3390/antibiotics11050546

**Published:** 2022-04-20

**Authors:** Julie Gorham, Fabio Silvio Taccone, Maya Hites

**Affiliations:** 1Department of Intensive Care, Hôpitaux Universitaires de Bruxelles (HUB)-Erasme, Université Libre de Bruxelles, 1070 Brussels, Belgium; fabio.taccone@erasme.ulb.ac.be; 2Clinic of Infectious Diseases, HUB-Erasme, Université Libre de Bruxelles, 1070 Brussels, Belgium; maya.hites@erasme.ulb.ac.be

**Keywords:** ceftazidime–avibactam, meropenem–vaborbactam, imipenem–relebactam, ceftolozane–tazobactam, cefiderocol, eravacycline, acute kidney injury, renal replacement therapy

## Abstract

There is currently an increase in the emergence of multidrug-resistant bacteria (MDR) worldwide, requiring the development of novel antibiotics. However, it is not only the choice of antibiotic that is important in treating an infection; the drug regimen also deserves special attention to avoid underdosing and excessive concentrations. Critically ill patients often have marked variation in renal function, ranging from augmented renal clearance (ARC), defined as a measured creatinine clearance (CrCL) ≥ 130 mL/min*1.73 m^2^, to acute kidney injury (AKI), eventually requiring renal replacement therapy (RRT), which can affect antibiotic exposure. All novel beta-lactam (BLs) and/or beta-lactam/beta-lactamases inhibitors (BL/BLIs) antibiotics have specific pharmacokinetic properties, such as hydrophilicity, low plasma–protein binding, small volume of distribution, low molecular weight, and predominant renal clearance, which require adaptation of dosage regimens in the presence of abnormal renal function or RRT. However, there are limited data on the topic. The aim of this review was therefore to summarize available PK studies on these novel antibiotics performed in patients with ARC or AKI, or requiring RRT, in order to provide a practical approach to guide clinicians in the choice of the best dosage regimens in critically ill patients.

## 1. Introduction

Bacterial resistance to antibiotics is a major public health problem. There is currently an increase in the emergence of multidrug-resistant strains that have become endemic in Europe and many other areas of the world [1,2,3]. Bacterial resistance to β-lactams is mainly based on two mechanisms: the production of penicillin-binding proteins in Gram-positive bacteria and the production of β-lactamases in Gram-negative bacteria. In particular, Ambler’s classification [4] distinguishes four groups of β-lactamases (Table 1): the β-lactamases belonging to class B (metallo-β-lactamases) are very different from classes A, C, and D.

Multidrug-resistant bacteria (MDR) are bacteria resistant to at least one antibiotic molecule belonging to more than three different classes among the ones usually active on this strain. These are mainly methicillin-resistant *Staphylococcus aureus* (MRSA), extended-spectrum beta-lactamase-producing (ESBL) *Enterobacteriaceae*, and non-fermenting Gram-negative bacilli (*Pseudomonas aeruginosa*, *Acinetobacter baumannii*), multi-resistant to antibiotics. Infections caused by these bacteria are significantly associated with an increased mortality risk [5,6]. As such, in recent years, novel antibiotics have been developed to reduce the impact of MDR Gram-negative [1] and Gram-positive infections [7]. If the choice of the molecule is important when treating an infection (so that the drug chosen has in vitro activity against the infecting pathogen), then drug regimens also deserve special attention. This is because insufficient drug concentrations (i.e., underdosing), either in the blood or the target tissue, according to pharmacokinetic/pharmacodynamic (PK/PD) targets, can result in therapeutic failure or promote further resistance development, while excessive drug levels may cause toxicity [8,9].

Wide variability in renal function, ranging from augmented renal clearance (ARC), defined as a measured creatinine clearance (CrCL) ≥ 130 mL/min*1.73 m^2^, to acute kidney injury (AKI), eventually requiring renal replacement therapy (RRT), are often observed in critically ill patients and may affect drug exposure in these patients [10]. All novel beta-lactam (BLs) and/or beta-lactams/beta-lactamases inhibitors (BL/BLIs) antibiotics have specific properties, such as hydrophilicity, low plasma–protein binding, small volume of distribution, low molecular weight, and predominant renal clearance (Table 2) [8], which require daily dose adjustment in the presence of abnormal CrCL values or RRT use [11]. Moreover, this is a frequent clinical challenge, as AKI occurs in up to 50% of critically ill patients [12], with 5–10% of them eventually requiring RRT during their hospital stay [13]. Additionally, RRT can be administrated either as intermittent (i.e., peritoneal dialysis and intermittent haemodialysis) or continuous (i.e., continuous venovenous hemofiltration—CVVH; continuous venovenous haemodialysis—CVVHD; continuous venovenous hemodiafiltration—CVVHD) techniques [14]. Additional hybrid methods also exist, such as sustained low-efficiency dialysis and slow continuous ultrafiltration, which attempt to combine the benefits of both intermittent and continuous methods. The fundamental aim of all RRT techniques is the removal of unwanted solutes and water through a semipermeable membrane, using the processes of diffusion, convection, or a combination of both. Haemodialysis uses the principle of diffusion, which allows the transport of solutes across a semipermeable membrane generated by a concentration gradient; diffusion is more effective at removing low-molecular-weight molecules (<30 kDa), such as urea, creatinine, ions, and ammonia. Hemofiltration uses convection as the mechanism for blood detoxification; convection refers to the flow of solutes across a semipermeable membrane together with solvents, in a way that is dependent on the hydrostatic pressure gradient and membrane characteristics. The movement of solutes across the semipermeable membrane is also dependent on both the size of the molecule in question, the size of the pores in the membrane, the degree of protein binding, and whether the substance is water-soluble or not.

Drug PKs in patients on intermittent or continuous RRT (CRRT) is complex and depends on the modality used, the ultrafiltrate and dialysate flow rates, the type of semipermeable membrane, pre- vs. post-dilution mode, the degree of protein binding of the drug, and the disease state of the patient [21,22]. As such, standard doses of antimicrobials may result in underdosing or excessive concentrations [22,23].

The aim of this mini review was therefore to review the existing literature and guide clinicians in the choice of the best daily regimens of these novel antibiotics (including a new tetracycline) in critically ill patients with MDR infections and altered renal function. This particularly concerns patients with ARC or AKI, or those requiring continuous RRT.

## 2. Methods

A literature search was conducted on PubMed/MEDLINE from inception to February 2022, in order to retrieve prospective or retrospective studies, case series/reports, and clinical trials concerning the use and dosing adjustments of novel agents in critically ill renal populations (namely patients affected by AKI, requiring RRT, or exhibiting ARC). The antibiotics ceftazidime–avibactam, meropenem–vaborbactam, imipenem–relebactam, ceftolozane–tazobactam, cefiderocol, and eravacycline were included, and the following search was specifically created: (“ceftazidime–avibactam” OR “meropemen-vaborbactam” OR “imipenem–relebactam” OR “ceftolozane–tazobactam” OR “cefiderocol” OR “eravacycline”) AND (“acute kidney injury” OR “renal replacement therapy” OR “continuous renal replacement therapy” OR “hemofiltration” OR “hemodiafiltration” OR “haemodialysis” OR “continuous venovenous haemodialysis” OR “continuous venovenous hemofiltration” OR “continuous venovenous hemodiafiltration” OR “augmented renal clearance” OR “therapeutic drug monitoring” OR “dosage”). Articles investigating novel antibiotics in non-critically ill patients or without acute renal failure were excluded. Only articles published in English were included. Of the 105 articles found, we excluded 46 articles. No additional analyses on the risk of bias of each study, or a meta-analysis of the existing data was performed, as the available literature is limited, and the intention of this review was purely descriptive.

## 3. Results

### 3.1. Summary of Available Data

Treatment indications for these novel antibiotics were mainly complicated intra-abdominal infections, complicated urinary tract infections, and hospital-acquired pneumonia (HAP), including ventilator-associated pneumonia (VAP). For novel antibiotics, which have a bactericidal activity dependent on the duration of exposure, the PK/PD index that best correlates with the PD effect is the fraction of time for which the free drug concentration in plasma exceeds the minimum inhibitory concentration of the infecting microorganism over the dosing interval (*f*T > MIC) [8]. Although robust clinical data are lacking, a recent consensus paper proposed a *f*T > MIC > 100% as the optimal target in this setting [24]. Concerning eravacycline, the PK/PD parameter that demonstrated the best correlation with antibacterial response is the ratio of area under the plasma concentration–time curve over 24 h to the minimum inhibitory concentration (AUC0-24h/MIC) [25]. 

Multiple daily dosing coupled with prolonged infusion (i.e., extended or continuous infusion) may represent the best approach to maximize the time-dependent antimicrobial activity of beta-lactam antibiotics [26]. Among the novel BL and/or BL/BLIs, only meropenem–vaborbactam and cefiderocol (extended infusion in 3 h) and ceftazidime–avibactam (extended infusion in 2 h) were evaluated using this rationale. Conversely, both ceftolozane–tazobactam (infusion in 1 h) and imipenem–relebactam (infusion in 30 min) were only evaluated using the conventional intermittent infusion. Moreover, no higher than recommended loading doses, given as a 30 or 60 min bolus, have been proposed and/ or evaluated, despite data suggesting the need for such doses when administering hydrophilic antibiotics to critically ill patients [27].

In patients with AKI or RRT, a decrease in the grams per dose while maintaining unchanged the dosing interval appeared to be the most appropriate strategy for optimizing PK/PD target attainment of these drugs [28]. However, as rapid changes in CrCL can be observed in these patients, with prompt recovery of renal function—which generally occurs within the first 48 h after the initiation of hemodynamic resuscitation and other supportive treatment [29,30]—novel BL or BL/BLIs antibiotics should be administered using standard drug regimens, unadjusted for renal dysfunction, during the first 48 h, because transient AKI could result in drug underdosing, treatment failure, and poor clinical outcomes when lower daily doses are given [29]. Adjustment of drug regimens to renal function should be applied only after 48 h for patients with persistent AKI [23,29]. Importantly, most recommendations for drug regimens are based on the administration of antibiotics when delivering “standard” CRRT doses (i.e., 20–25 mL/Kg*h); as such, whether these antibiotic dosage regimens remain effective in case of higher CRRT doses is unknown.

Altered dosing strategies based on continuous and prolonged infusions or a more frequent dose may represent a valuable approach for achieving optimal PK/PD target with novel BL and/or BL/BLIs in ARC patients [31,32].

### 3.2. Ceftazidime–Avibactam

Ceftazidime–avibactam is the association between ceftazidime and a new beta-lactamase inhibitor, avibactam. Avibactam inhibits most β-lactamases, i.e., those belonging to Ambler’s classes A, C, and D. In particular, avibactam effectively inhibits *Klebsiella pneumoniae* carbapenemases (KPC) and *Enterobacteriaceae* spp. oxacillinase (OXA-48)-producing strains [33,34,35]. On the other hand, avibactam has no activity on class B enzymes (metallo-β-lactamases) or certain OXAs produced by *Acinetobacter* spp., and has limited activity on anaerobes [36]. The standard ceftazidime–avibactam dosage regimen is 2 g/500 mg every 8 h, given as a 2 h infusion.

This antibiotic is predominately eliminated via the kidney, and doses must be adjusted according to renal function (Figure 1) [37,38]. Indeed, a phase I study [39] demonstrated that avibactam total plasma CL decreased with increasing severity of renal impairment and that avibactam was extensively removed by haemodialysis. However, decreased drug efficacy was observed in a phase III clinical trial in patients with moderate renal impairment for complicated intra-abdominal infections [40], potentially as a result of rapidly improving renal function during therapy with consequent underdosing due to reduced daily regimens. Moreover, in patients with moderate renal impairment (CrCL 31–50 mL/min), the study also found an increased clinical cure in the meropenem arm (1000 mg q12h) when compared with the ceftazidime–avibactam arm (1000/250 mg q12h; 45.2% vs. 74.3%, *p* = 0.016). However, caution in interpreting these findings is required. Patients treated with ceftazidime–avibactam had a greater relative reduction in the dose when compared with those receiving meropenem (66% vs. 33%). As such, this study led to increase the recommended dose of ceftazidime–avibactam in patients with moderate renal impairment from 1000/250 mg q12h to 1000/250 mg q8h [41,42,43]. This modified dosage adjustment was validated in another phase III trial [43]. These results are in contrast with PK/PD modelling analyses suggesting that both strategies (i.e., 1000/250 mg q12h or 1000/250 mg q8h) could achieve a PK/PD target of 50% *f*T > MIC against *Klebsiella pneumoniae* carbapenemases-producing *Klebsiella pneumoniae* in patients with moderate renal impairment [16]. A CrCL ≤ 50 mL/min was the only covariate identified to warrant dose adjustments [44].

Two case reports assessed the PK characteristics of ceftazidime–avibactam in critically ill patients requiring CRRT [45,46]. Wenzler et al. [46] found that ceftazidime–avibactam administered as 1000/250 mg q8h in a 2 h infusion achieved optimal drug concentrations (100% *f*T > MIC). Conversely, Soukup et al. [45] found that only standard drug regimens (i.e., 2000/500 mg q8h) could achieve high trough concentrations (i.e., >32 mg/L) during CVVHDF. Moreover, the use of RRT was a predictor of the development of resistance in a study of 77 patients treated with ceftazidime–avibactam for carbapenem-resistant *Enterobacteriacae* spp. [47]. As such, in critically ill patients receiving CRRT, drug regimen of 1000/250 mg q8h can be used for more susceptible strains (i.e., MIC < 4 mg/L), while higher regimens should be considered for less susceptible strains.

A subgroup analysis of 239 patients with ARC included in the ceftazidime–avibactam versus meropenem trial in patients with nosocomial pneumonia (REPROVE) showed that the standard dosage of 2000/500 mg q8h as a 2 h infusion ensured a >95% probability of target attainment of 50% *f*T > MIC for MICs up to 16 mg/L [43]. Population PK models have also confirmed that dose adjustments were not warranted for patients with ARC [44].

### 3.3. Meropenem–Vaborbactam

Meropenem–vaborbactam is the combination of a carbapenem with a new beta-lactamase inhibitor, vaborbactam. Vaborbactam is the first boronic acid class β-lactamase inhibitor. It inhibits class A and C β-lactamases, and is particularly active against KPC [48,49], while it is not active on class D enzymes, such as OXA-48 [49]. Although meropenem–vaborbactam is active against *Pseudomonas aeruginosa*, this appears to be similar to that of meropenem alone [33,49]. Meropenem–vaborbactam is prescribed at a dose of 2000/2000 mg q8h as a 3 h infusion.

Its elimination is renal and doses must be adapted to renal function (Figure 1) [19,50,51]. Indeed, the PKs of meropenem–vaborbactam at 1000/1000 mg was evaluated in a phase I, open-label study including 41 subjects with chronic renal impairment [19]. This study showed that plasma clearance of meropenem and vaborbactam was progressively reduced with decreases in CrCL, justifying the need for dose adjustment in this situation. On the other hand, haemodialysis effectively removed both drugs, justifying the need for a second dose after the dialysis session [19,52].

One case report assessed the PK characteristics of meropenem–vaborbactam in one anuric critically ill patient undergoing CRRT [53]. Both 1000/1000 mg and even lower doses given over 3 h allowed the achievement of a PK/PD target above the MIC of the isolated strain. No data on meropenem–vaborbactam concentrations in patients with ARC are currently available.

### 3.4. Imipenem–Relebactam

Imipenem–relebactam is a combination of a carbapenem and a new beta-lactamase inhibitor, relebactam (to which is added cilastine, which limits the renal metabolism of imipenem) [54]. This drug has demonstrated activity against MDR strains of *P. aeruginosa* and many carbapenemases-producing *Klebisella* or *Enterobacteriaceae* strains in in vitro studies [55,56]. Relebactam is structurally close to avibactam, but it is less active against class D β-lactamases [55]. Imipenem–cilastine–relebactam is prescribed at a dose of 500 mg/500 mg/250 mg q6h as a 30 min infusion.

Drug elimination is renal and the doses must be adapted to the renal function (Figure 1) [18]. Currently, there are no published PK data in critically ill patients with AKI or receiving CRRT. However, imipenem–relebactam clearance was assessed in ex vivo continuous hemofiltration (CH) and continuous haemodialysis (CHD) models, showing that imipenem and relebactam are not removed by adsorption, but effectively cross the hemodiafiltration membrane during CH and CHD [57]. In this setting, a dose of 200 mg/100 mg q6h was sufficient to achieve PK/PD targets. A warning about the risk of insufficient drug concentrations when CrCL exceeds 150 mL/min is found on the drug label of imipenem–relebactam, but without a proposal for dose adjustments.

### 3.5. Ceftolozane–Tazobactam

Ceftolozane–tazobactam is the combination of a new cephalosporin, ceftolozane, and a well-known beta-lactamase inhibitor, tazobactam. Ceftolozane has anti-*Pseudomonas* activity, superior to other drugs (i.e., ceftazidime, carbapenems, ciprofloxacin) [58,59]. This drug has reduced activity on multi-resistant *Enterobacteriaceae* when compared with ceftazidime–avibactam, and has poor activity on anaerobes and Gram-positive bacteria [60]. The recommended dosage is 1000/500 mg q8h as a 1 h infusion. In case of severe infection, the dosage should be increased to 2000/1000 mg q8h.

Drug elimination is renal and the doses have to adapted to the renal function (Figure 1) [17]. A phase I study demonstrated that plasma tazobactam concentrations increased with decreasing CrCL, and that over 50% of the administered drug was removed during a 4 h haemodialysis session [61]. Monte Carlo simulations were conducted and confirmed that dosing regimens approved by the Food and Drug Administration (FDA) (i.e., 1000/500 mg in patients with ARC and CrCL > 50 mL/min; 500/250 mg if CrCL 30–50 mL/min; 250/125 mg in patients with CrCL < 30 mL/min; 500/250 mg followed by a maintenance dose of 100/50 mg q12h with dialysis) were sufficient to achieve target drug levels to ensure bactericidal activity at all the selected MIC breakpoints [62].

Several case reports and population PK studies have assessed the PK characteristics of ceftolozane–tazobactam in critically ill patients requiring CRRT, with different dosing schedules and modes of administration. An ex vivo study showed that CL of ceftolozane–tazobactam mainly depended on effluent flow rate and that a dose of 500/250 mg to 1000/500 mg q8h met adequate drug levels at contemporary effluent rates [63]. Gatti et al. [64] confirmed the significant correlation between effluent flow rate and ceftolozane CL. Sime et al. [65] also showed, in critically ill patients receiving CVVHDF with an effluent flow rate of <2.5 L/h, that a single 2000/1000 mg loading dose followed by 500/250 mg q8h could achieve a 40% *f*T > MIC target against *Pseudomonas aeruginosa*. Considering a higher target of 100% *f*T > MIC, a daily dose of at least 1000/500 mg q8h was required. Aguilar et al. [66] reported in a patient undergoing CVVHD with an effluent flow rate of 3 L/h, that an intermittent infusion of ceftolozane–tazobactam (2000/1000 mg q8h) resulted in optimal PK/PD target. Mahmoud et al. [67] also found that a similar ceftolozane–tazobactam regimen (2000/1000 mg q8h) reached optimal drug levels in a critically obese patient requiring CVVHDF with a high effluent flow rate (4 L/h). Other studies have been carried out in patients treated with CVVH or CVVHDF with a low effluent flow rate but with highly adsorptive membranes and have shown that more restrictive PK/PD targets (i.e., 100% *f*T > MIC) were achieved with high doses and prolonged infusion (i.e., 2000/1000 mg q8h using an extended-infusion time of 4 h) [68,69,70,71].

In a population PK study by Sime et al. [72], conducted in patients with ARC, an intermittent infusion of 1000/500 mg q8h achieved the PK/PD target of 40–60% *f*T > MIC, for a susceptibility threshold of 8 mg/L. When considering a more aggressive target of 100% *f*T > MIC, a dosage of 2000/1000 mg q8h was required, with still insufficient drug levels in some patients. However, the use of continuous infusion regimens (1000/500 mg q8h loading dose followed by a 3000/1500 mg continuous infusion) was adequate to achieve this more restrictive target in all patients. Conversely, Nicolau et al. [73] recently showed that in 11 critically ill patients with ARC (i.e., median CrCL of 214 mL/min), a single dose of 2000/1000 mg q8h resulted in unbound ceftolozane levels above 4 mg/L for approximately 6 h in 9 (81.8%) patients and for up to 8 h in 7 (63.6%) patients. Unbound tazobactam levels remained > 1 mg/L for 2 h in all patients and for up to 4 h in 7 patients (63.6%); as such, a dose of 2000/100 mg q8h should provide sufficient concentrations in patients with ARC, without additional dose adjustments. This was also suggested by a recent randomized trial, using a PK/PD target of 50% *f*T >MIC of 4 mg/L against *Pseudomonas aeruginosa* [74]. In case of resistant strains (i.e., MIC ranging from 8 to 32 mg/L), a prolonged infusion of 2000/100 mg q8h over 4 h was associated with better probability of target attainment compared with intermittent infusions [75].

### 3.6. Cefiderocol

Cefiderocol is a cephalosporin that has a new mechanism of action: one of the side chains has been replaced by a siderophore, allowing it to enter bacteria using iron channels [76]. The presence of the siderophore protects it from the hydrolysis of almost all beta-lactamases and carbapenemases, including the metallo-beta-lactamase type B [77,78]. It is also active on *Acinetobacter* spp., resistant to imipenem, and on pan-resistant strains of *Pseudomonas aeruginosa* [77]. It appears to also have activity against *Stenotrophomonas maltophilia* and *Burholderia cepacian* [79]. The molecule is administered intravenously at a dose of 2000 mg q8h, as a 3 h infusion.

Cefiderocol is mainly eliminated by the kidneys [15]. Katsubi et al. [80] showed that cefiderocol exposure increased in subjects with moderate and severe renal impairment or on RRT, when compared with those with normal renal function. Therefore, PK/PD modelling and simulation were conducted to determine dose adjustment based on renal function (Figure 1) [32]. In one study [80], approximately 60% of cefiderocol was removed by a 3–4 h haemodialysis (HD) session, suggesting the need for a supplemental dose in patients receiving this therapy.

Cefiderocol is the first of these new antibiotics for which dosing recommendations are reported in the summary of product characteristics for patients undergoing CRRT. The dosage was calculated on the basis of cefepime clearance by CRRT, considering that these two BLs share similar features in terms of molecular weights and protein binding [81]. The recommended dosing of 1000 mg q12h over a 3 h infusion during CVVH and 1500 mg q12h over a 3 h infusion during CVVHD and CVVHDF, with 90% target attainment against pathogens with an MIC ≤ 4 mg/L [81]. However, one case report assessed the PKs of cediferocol in a patient receiving CVVHDF and suggested the need for a higher drug dose (2000 mg q8h as a 3 h infusion) to be effective against strains with a MIC ≤ 8 mg/L.

A more aggressive dosage of 2000 mg q6h administered by extended infusion over 3 h is currently recommended for patients with ARC, based on Monte Carlo simulations performed on data obtained from phase III clinical trials, aiming for a >90% probability of target attainment 75% *f*T >MIC for strains with a MIC of ≤4 mg/L [32].

### 3.7. Eravacycline

Eravacycline is a fluorocycline, whose structure and PK characteristics are close to those of tigecycline [82]. This drug is poorly eliminated renally. It has a broad spectrum of activity, covering major Gram-positive bacteria (streptococci, vancomycin-resistant *Staphylococcus aureus*, and *Enterococcus* spp., including glycopeptide-resistant isolates) and most Gram-negative bacteria [83]. It is active against most anaerobes, and certain strains of *Acinetobacter baumannii*, but also *Legionella pneumophila* and *Francisella tularensis* [20]. The notable exception in this spectrum is *P. aeruginosa*. Its activity is unaffected by most mechanisms conferring resistance to other tetracyclines [82]. Eravacycline is a substrate of the cytochrome P450 isoenzyme CYP3A4. The daily dosage varies between 1.0 and 1.5 mg/kg divided in one or two injections, with the drug administered over 1 h.

Following a single intravenous administration, circulating drug levels in subjects with RRT and mild or moderate renal impairment were similar to those observed in healthy subjects. Therefore, no dose adjustment should be required in patients with abnormal renal function [84].

## 4. Conclusions

Novel antibiotics have recently been approved and should progressively be added to the therapeutic arsenal against infections due to multidrug-resistant bacteria. Despite the fact that critically ill patients are at the highest risk of presenting these infections, and that a great majority of them have abnormal CrCL, scarce data are currently available on optimal dosage regimens in this setting. However, insufficient drug levels might be observed for these drugs during critical illness, which might increase the risk of therapeutic failure or emergence of resistances. Therapeutic drug monitoring (TDM) may represent a helpful tool to adjust these novel BLs and/or BL/BLIs; however, this is currently not routinely available. Dose adjustments for these antibiotics have been summarized in Figure 1 based on current extant data. Importantly, the presence of additional risk factors for underdosing (i.e., obesity, high effluent rates during CRRT, high MICs) should also be considered during drug prescription and might influence drug regimens.

## Figures and Tables

**Figure 1 antibiotics-11-00546-f001:**
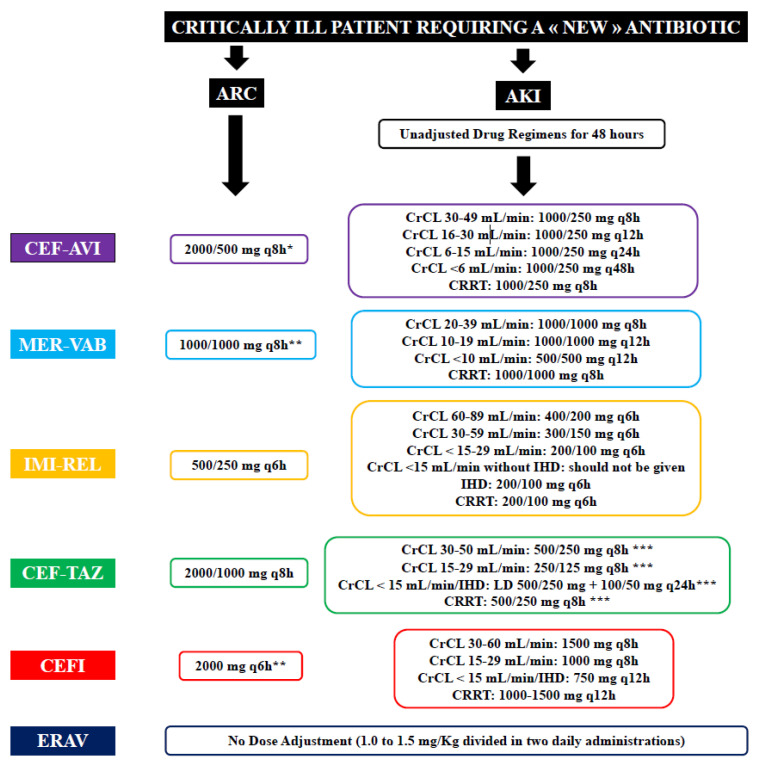
Summary of dose adjustment for patients with augmented renal clearance (ARC) or acute kidney injury (AKI). CEF-AVI—ceftazidime–avibactam; MER-VAB—meropenem–vaborbactam; IMI-REL—imipenem–relebactam; CEF-TAZ—ceftolozane–tazobactam; CEFI—cefiderocol; EVAR—Eravacycline; CrCL—creatinine clearance; IHD—intermittent haemodialysis; CRRT—continuous renal replacement therapy. *—2 h infusion; **—3 h infusion; ***—dosage recommendation for complicated intra-abdominal and urinary tract infections; for hospital-acquired pneumonia, dose should be doubled.

**Table 1 antibiotics-11-00546-t001:** Ambler’s classification of beta-lactamases (BL).

Class A Non-Metal (Serine)	Class B Metal (Zinc)	Class C Non-Metal (Serine)	Class D Non-Metal (Serine)
Classical narrow spectrum Extended-spectrum beta-lactamases (ESBL) Class A carbapenenemases	Metallo beta-lactamase	AmpC beta-lactamase Extended-spectrum AmpC	Oxacillinase Carbapenem-hydrolysing class D beta-lactamase

**Table 2 antibiotics-11-00546-t002:** Main pharmacokinetic properties of novel beta-lactam (BLs) and/or beta-lactams/beta-lactamases inhibitors (BL/BLIs) and tetracycline.

Drug	VD, L	T_1/2_, H	Protein Bound, %	Renal CL
Cefiderocol [15]	13.5/26.6	2–3	40–60	90–98%
Ceftazidime–avibactam [16]	17.0/22.2	1.5–2.7	7–10	72–87%
Ceftolozane–tazobactam [17]	13.5/18.2	3.1	16–30	62–84%
Imipenem–relebactam [18]	19.0/24.3	1.2	20–22	52–92%
Meropenem–vaborbactam [19]	18.6/20.2	2.3	2–33	74%
Eravacycline [20]	321	24	80–90	34%

VD—volume of distribution; T_1/2_—half-life; CL—clearance.

## Data Availability

Not applicable.

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
