# Peer review of "Drug Regimens of Novel Antibiotics in Critically Ill Patients with Varying Renal Functions: A Rapid Review"

_antibiotics, 2022, doi:10.3390/antibiotics11050546_

Round 1

Reviewer 1 Report

Antimicrobial Stewardship is extremely critical in preventing antimicrobial resistance. Similarly adverse effects, most notably renal failure from use of antimicrobials is a known entity and this is made even more difficult when novel antimicrobials are introduced into the system. I think the authors have done an extensive research of literature to discuss novel antimicrobials and studied the novel antimicrobials in the setting of renal impairment and does adjustment based upon the pk/pd. 

Overall the background statement and the literature search is excellent. the language overall is also easy to understand. As a reviewer and an educator , I would have preferred if the Figure 1 can be added on page 2 so that the reader can get a sense of the antimicrobial they plan to use. 

Author Response

Thank you for your suggestion. We have made changes in the manuscript.

Reviewer 2 Report

Methods - it is necessary to specify the year of search - criteria for excluding articles from the search - how many articles were reviewed and how many did not meet the search criteria Conclusions. - It would be nice to get some recommendations on the use of antibiotics and renal failure (at the discretion of the editor and authors)

Conclusions. - It would be nice to get some recommendations on the use of antibiotics and renal failure (at the discretion of the editor and authors)

Author Response

We thank the reviewer for the proposal - as recommendations are quite complex, we prefer to refer to the main Table and Figure of the manuscript and not in the Conclusions section.

Reviewer 3 Report

Julie Gorham et and co-workers reviewed the drug regimens of the new antibiotics in critically ill patients. On the whole, the review is appeared to be interesting, updated and the methodologies are appropriate. The authors discussed their findings well and supported their view with substantial works of literature.  

I have no major comments on this manuscript.  However, the authors are needed give more explanation on the dosage for patients on CRRT, and underline it.

In detail, I suggest:

  • adding a third column in Figure 1 with dosage in patients on RRT, to give greater prominence to patients with dialytic-AKI
  • precising whether the dosage adjustment for CRRT patients is intended at the recommended dialysis dose of 20-25 ml/Kg/hour

Author Response

We thank the reviewer about the recommendations on CRRT - we have included existing evidence in the text and added a recommendation into the Figure, whenever possible. However, we avoided to prepare a third column as the Figure will become too large and difficult to read. We have added a comment into the text regarding CRRT dose, as requested.